# Signal theory based encryption of faster-than-Nyquist signals for fiber and wireless transmission
Abhinand Venugopalan ✉, Karanveer Singh, Janosch Meier & Thomas Schneider

New applications such as the Internet of Things, autonomous driving, Industry X.0 and many more will transmit sensitive information via fibers and over the air with envisioned data rates beyond terabits per second. Therefore, the encryption has to be simple, fast and spectrally efficient, so that the power consumption and latency are low and the scarce bandwidth is not wasted. Various encryption schemes, based on mathematical algorithms, quantum theory, chaos communication or spectral spreading below the noise level have been explored. Besides power, spectral efficiency and latency, most of these approaches face additional challenges such as limited data rates, compatibility issues with communication standards and integration. Here, we propose a signal theory based method that enables the encryption of super-signals with bandwidths of hundreds of gigahertz without any additional bandwidth. In proof-of-concept experiments we demonstrate the encryption of a 270 GBd faster than Nyquist super-signal in a 252.4 GHz bandwidth. The encryption is simple, fast and power efficient, and offers a solution for secure data transmission in existing and future communication networks.

The exponential growth of communication data traffic has led to a heightened reliance on optical networks and wireless infrastructure across a spectrum of applications such as information exchange, autonomous driving, public health care, infrastructure monitoring, and more[1–3]. With more than 5.3 billion active internet users around the globe, representing 65.4% of the world's population, the exchange of data over the network has created the need for secure transmission of sensitive information between end users[4]. To secure the communication, the content of the message can be converted into an encrypted form so that the confidentiality of the transmitted information is maintained during data transmission. At present, the most commonly used encryption systems are based on computational security[5,6]. In the seven-layer open system interconnection (OSI) model, these security measures are often implemented at a higher layer by digital data encryption[7]. However, at any lower level of the OSI model and especially at the physical layer where raw bit streams are transmitted, the encryption is still vulnerable, as an eavesdropper can make copies of the signals[8]. The unauthorized user can then attempt to break the digital encryption by a massive post-processing of the stored data. Hence the strength of the security lies only in the computational difficulty of breaking the encryption algorithm. In the past, it was considered that the encryption cannot be broken in a finite time-frame. However, the latest developments in parallel high-performance computer networks and quantum computers may challenge this statement[6,9].

In order to secure data transmission, several methods have been proposed in recent years that guarantee security directly on the physical layer, known as Physical Layer Security (PLS)[10–17]. In PLS, the data is either mapped to noise-like signals, such that the eavesdropper cannot detect the presence of any transmitted bit stream in the channel, or through measures where an attempt of interception is alerted to the authorized end users. The latter is for instance provided by quantum communication[18] or quantum key distribution (QKD), where any process of copying is readily noticed according to the properties of the no cloning theorem[19]. In its present state, however, QKD has shown technical limitations, since it requires special equipment, cannot be transmitted over very long distances and the generation and detection of quantum states is inherently slow[20], with a maximum reported rate of 2.5 GHz[21].

The encryption into noise-like signals has been mainly based on masking the encrypted signal such that an unauthorized access results in the detection of no meaningful information for post processing. One such method is steganography, where the signal spectrum is spread below the noise level of the channel by a dispersive element, for instance. Though it provides an additional layer of protection, signal security is not ensured as a tunable dispersion-compensating device can indicate the presence of any hidden signal[22].

Another method of spreading the spectrum is code division multiple access (CDMA), where the signal is encoded with a specific code in the time

THz-Photonics Group, Technische Universität Braunschweig, Braunschweig, Germany. ✉e-mail: abhinand.venugopalan@tu-braunschweig.de

or wavelength domain[17]. For instance, a 40 Gbit/s data transmission using QKD-assisted optical CDMA has been demonstrated[12]. Another possibility is spreading the signal energy beyond the detector bandwidth. A 10 GBd QPSK signal was spread over 200 GHz by an optical frequency comb using this method[16]. The main problem of all spread spectrum methods, however, is that they need much more bandwidth than required for the signal transmission. The spectrum of a 40 GBd signal in a 50 GHz grid dense wavelength division multiplexing (DWDM) network, for instance, cannot simply be spread, since this negatively affects the transmission of the neighboring channels.

Another approach is chaos based communication, where a broadband chaotic signal is used as an optical carrier[13]. A 50 Gbit/s wavelength multiplexed system using chaotic phase scrambling has been experimentally demonstrated[14]. The method is highly sensitive to the initial hardware conditions and chaos signal synchronization between the transmitter and receiver. The encrypted signal also tends to occupy at least twice the original spectrum and in general requires long processing times[23].

Secure current and future communication requires an easy, on-the-fly, adaptable solution along with the compatibility to handle terabits per second transmission rates that can be en- and decrypted with low-power, low-footprint integrated devices. Here, a real-time encryption method based on the fundamentals of signal theory is proposed. The encrypted signal does not need any additional bandwidth, and the encryption can be easily implemented in any communication system, either through hardware or software processing, suitable for the next-generation fiber and wireless infrastructure. The encryption method is completely transparent to any modulation format and even orthogonal frequency division multiplexed (OFDM), analog radio over fiber signals, or signals already encrypted at a higher OSI level, can be encrypted. Furthermore, the encryption can be changed during transmission by a modulation with high speed. In a proof-of concept, a faster-than Nyquist transmission of a 270 GBd signal in a bandwidth of 252.4 GHz is presented with low-speed electronics and photonics. No specific system adaptation with respect to the transmitted signal bandwidth or the data formats was required. Moreover, the encryption can be implemented in software, hardware, or in algorithms for digital signal processing.

## Results
### Principle of EN- and decryption

The presented concept of encryption is based on two fundamental ideas. The first is to encrypt and transmit super-signals[24], which are formed by combining multiple rectangular-bandwidth Nyquist signals, modulated onto a flexible carrier source. The bandwidth of these super-signals exceed that of state-of-the-art receivers, storage devices and digital signal processing, so that it cannot be received and stored for post-processing at once. The only possibility for an attack is to divide the real time, high-bandwidth super-signal into spectral slices by filters and electronic signal processing[25]. However, as we will show, to successfully decrypt the signal, or the signals, the bandwidth of these filters can differ by only $10^{-3}$% to that of the encrypted signal. Moreover, even if the entire signal spectrum could be reconstructed by spectral slicing, decryption is only possible by trying out the various key options for all possible center frequencies and bandwidths within the encrypted super-signal.

Since Nyquist signals are used, the super-signal can be transmitted with the maximum possible symbol rate and no bandwidth is wasted for the encryption. In our proof-of concept experiments, we will even show the en- and decryption of a 252.4 GHz super-signal with a 7% higher symbol rate than given by the Nyquist limit.

The second fundamental idea of the method is that each signal, limited to the baseband bandwidth $B/2$, can be fully represented by its orthogonal sampling points taken with a sampling rate of at least $B$. For the encryption we destroy that orthogonality so that the interference between the sampling points results in a noise-like signal for transmission, that cannot be detected without knowledge of the key. For the decryption the orthogonality is restored, bringing back the original signal.

Each signal of baseband bandwidth $B/2$ can be described as a superposition of orthogonal sinus cardinalis $\mathrm{sinc}(t) := \lim_{\substack{x \to t \\ x \in \mathbb{R} \setminus \{0\}}} \left( \frac{\sin(\pi x)}{\pi x} \right)$ pulses with the bandwidth and sampling rate $B$, weighted with the sampling points of that signal:

$$s(t) = \sum_{m=-\infty}^{\infty} s\left(\frac{m}{B}\right) \cdot \mathrm{sinc}\left(Bt - m\right) \tag{1}$$

Single sinc pulses, however, are unlimited in time and just a mathematical construct. But, each bandwidth-limited signal can be seen as the superposition of a number of $N$ sinc pulse sequences (SPS)[26–29]. As long as the sampling theorem is not violated, $N$ can be any odd number. The connection and differences between single sinc pulses and SPS are shown in Fig. 1a, b.

An SPS is the unlimited superposition of single unlimited sinc pulses which are time shifted to each other so that each single sinc pulse is in the zero crossings of all other sinc pulses. In the corresponding frequency domain this is a rectangular frequency comb with $n = (N-1)/2$ frequency lines plus the direct current, starting from zero and having the frequency separation $\Delta f$ to each other. After modulation, all $N$ lines will appear around the carrier and the SPS has the bandwidth $B = N\Delta f$. All frequency lines have the same or a linear-dependent phase. In the time domain, the sequences have $N-1$ zero crossings. The construction of a signal (black line) with SPS is shown in Fig. 1c. Here SPS with $N = 9$ have been chosen. In contrast to single sinc pulses, SPS define periodical sampling points but, as single sinc pulses they are orthogonal, if they are time shifted against each other so that the maxima of one are in the zero crossings of the others. Since the single SPS has nine zero crossings, eight more weighted SPS are needed to fill all zero crossings and to define all required sampling points for the signal. This summation corresponds to Eq. (1), so all sampling points of the signal and therefore the signal $s(t)$ itself is defined.

Suppose now that the phases of the frequencies in the comb are set to an arbitrary value. In this case, the frequency comb is no longer an SPS in the time domain and the orthogonality between the sampling points is destroyed. In the following we will refer to this as the "keyfunction". Figure 1d shows the signal after the same keyfunction has been applied to the frequencies of the nine-line SPSs. Due to the keyfunction, the phase of the frequency lines is not equal anymore, the orthogonality is destroyed and, due to intersymbol interference, the superposition is a kind of arbitrary, noise-like signal. Since no underlying structures are present, methods to recover a signal buried under noise cannot be used[30].

One possible implementation of the above idea is shown in Fig. 2. In the transmitter (left) the sub-samples in each of the $N$ branches are used to generate a low-bandwidth analog sub-signal, which is then multiplied with the time-shifted keyfunctions. The weighted keyfunctions are then added up and modulated on a carrier. On the receiver side (right), the inverse process is carried out to get back the original signal. This concept can be implemented in software, as an algorithm in a digital signal processor, as hardware or as a mixture between them.

With an additional comb source with $k$ frequency lines or a flexible multi-wavelength source this basic idea can be used to generate very broadband encrypted super-signals with the overall bandwidth $B_{SE} = \sum_{i=1}^{k} B_i$, as shown for optical signals in Fig. 3. For reaching the maximum spectral efficiency, each signal can be chosen to be a Nyquist data signal, so that the bandwidth shape of each of the signals is rectangular and corresponds to the symbol rate. Ideally, the symbol rates and bandwidths are different and the frequency spacing between any two adjacent lines of the flexible multi-wavelength source should be adapted to the different symbol rates, so that the spacing between the signals is unequal and no guardband between the signal spectra is measurable in the super-signal. The $k$ signals are encrypted (as described before with Fig. 2) with the same or different keyfunctions. The keyfunctions may differ in the key itself, but as well the

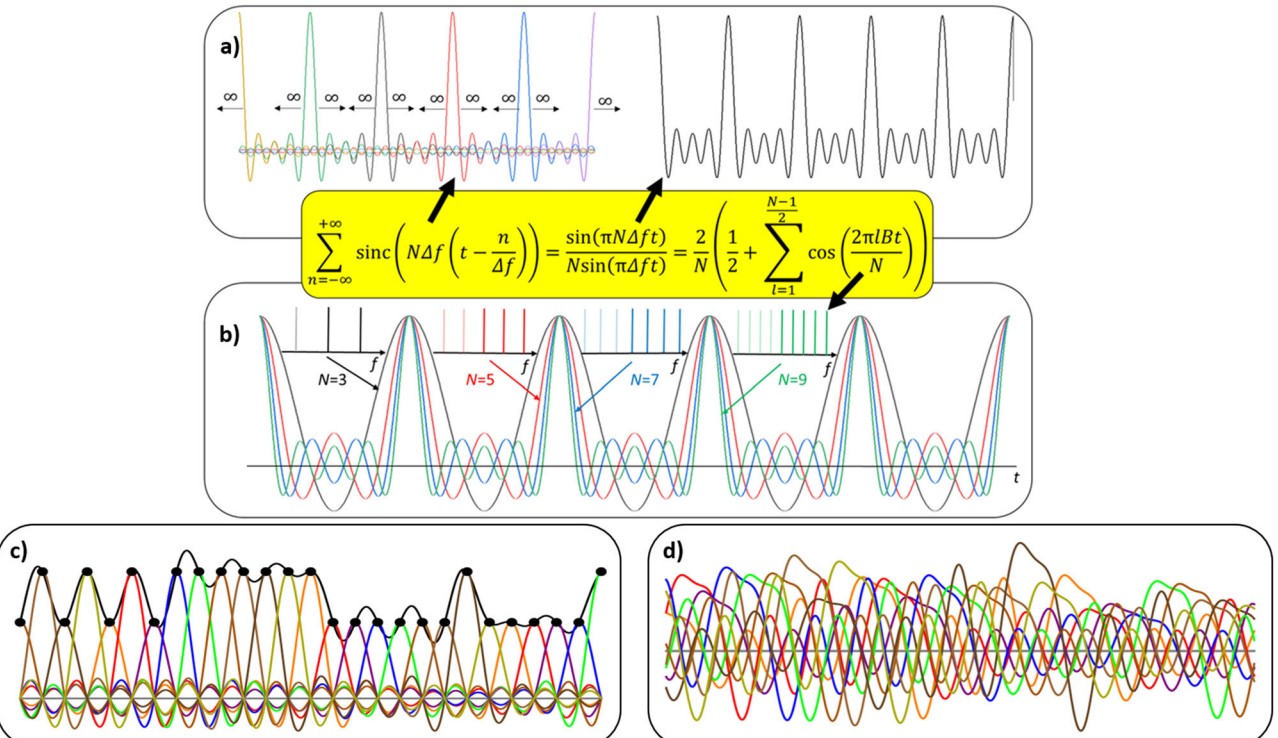

**Fig. 1 | Sinc pulse sequences and keyfunction.** Sinc pulse sequences (SPS) as the superposition of single, time-shifted ideal sinc pulses (**a**). In (**b**) SPS with a different number of comb lines $N$ are shown in the time and frequency domain. The $n = (N-1)/2$ thick lines correspond to the frequency combs for the baseband signal plus the direct current. After modulation on a carrier, the transparent lines on the left side will occur as lower sidebands. Therefore, only the baseband frequencies $n$ can be used for the keyfunction. The construction of a signal (black line) from nine time-shifted SPS ($N = 9$), periodically weighted with the sampling points of the signal is shown in (**c**), and (**d**) represents the nine former weighted SPS after the same phase key has been applied to their $n$ frequency lines in the baseband. Please note that only the superposition of all these lines can be detected.

key strength $N_i$ can be chosen arbitrarily. A Mach-Zehnder modulator (MZM) is incorporated to modulate the $k$ signals on one of the $k$ frequency lines and the signals are added up to build the super-signal. Since the spacing between adjacent frequency lines is the same, or slightly lower than the symbol rate (and thus bandwidth) of each signal, the result is a very broad rectangular spectrum without any guardband between the signals. As an example, an experimentally achieved spectrum of a super-signal is shown in Fig. 3a, please see the Methods section for details.

In the receivers for each of the sub-signals (Fig. 3b) the whole spectrum will be detected at once and power-split into a number of branches which correspond to the key strength $N_i$. In each single branch the whole spectrum is multiplied with the respective keyfunction. In the first set of experiments, this multiplication has been done with MZM, where the optical input is the whole signal spectrum and the electrical input is the keyfunction. This multiplication in parallel branches reduces the hardware complexity at the receiver. Since the sub-signal bandwidth is $N_i$-times lower than the signal bandwidth and $kN_i$ times lower than that of the super-signal, the signal to noise and distortion ratio and effective number of bit of the ADC is increased[29] and can in turn at least partly compensate for the increase in power consumption of the parallel signal processing. However, this means that the number of branches $N_i$ and therefore the key strength is fixed and defined by the hardware. Alternatively, the multiplication can be done in the baseband after detection and down-conversion, as done in the second series of experiments and described in Fig. 3c. In the case of multiplication in the baseband, the number $N_i$ and therefore the key strength can be very high and it can be varied, even during the transmission. A mixture between both approaches is possible as well. So for a keyfunction with $M \times N_i$ frequency lines, a number of $M$ hardware branches with an optical modulator for the multiplication of the received signal with an $M$-line

keyfunction reduces the bandwidth by $M$ for the following detection and electronic processing. For decrypting this signal, the additional $N_i$-line keyfunction can be applied in the software. This number $N_i$ is not fixed and can be changed.

The multiplication between the spectrum and the keyfunction in the time domain leads to a convolution between the spectrum of each sub-signal and the keyfunction in the spectral domain. The information of the sub-signal is in the spectral region of $B_i/N_i$ around the center of the convolution. This spectrum is then down-converted to the baseband by a coherent detector (CD) with that bandwidth. Alternatively, the bandwidth can be restricted to $B_i/(2N_i)$ by a software filter in the baseband. Please note that the baseband bandwidth ($B_i/2$) is half of the optical bandwidth. Since the sub-signal bandwidth is $N_i$-times lower than the signal bandwidth or the spacing between the optical carrier lines, no optical filtering is needed to separate the individual channels[31]. After analog to digital conversion in low bandwidth ADCs (again $B_i/(2N_i)$), the parallel samples can be used to restore the original signal.

The keyfunction may be changed with a speed up to the symbol rate of the signal by a phase modulation, as well the frequency spacing of the flexible carrier source and symbol rate of the single signals might be changed during transmission. However, at every time the keyfunction including bandwidth, as well as center frequency of the signal must be known by the receiver. Therefore, a secure channel is required for the transmission of this information. Since the parameters for the encryption have a low bandwidth and the secure channel has to exist for a short time only, secure information exchange via QKD, post quantum cryptography or even hardware security primitives can be considered for this.

If only a single low-bandwidth signal has to be encrypted, the security of the transmission may be further enhanced by filling a broadband rectangular spectrum with additional noise or with encrypted fake-signals.

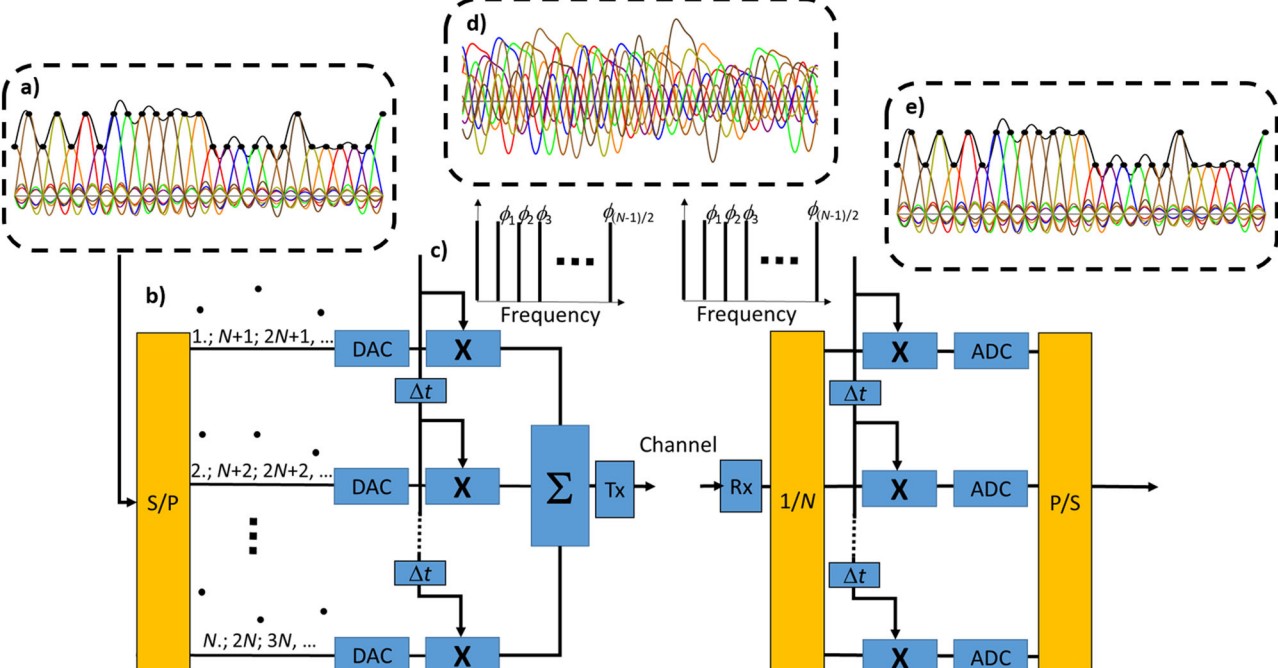

**Fig. 2 | Concept of encryption for the single signal.** In the transmitter (left), the incoming analog or digital signal $s(t)$ can be fully represented by its sampling points, as shown in the dashed box (**a**). In the first step these sampling points will be divided between the $N$ branches by a serial-to-parallel converter (S/P), so that in each branch the original sampling rate $R_s$ is reduced to $R_s/N$ represented by the sub-samples shown in (**b**). In each branch a low-bandwidth digital to analog converter (DAC) generates an analog sub-signal with the bandwidth $B/N$. Without applying a phase key, the $n = (N-1)/2$ electrical frequencies (**c**) would represent an SPS with $N$ frequency lines after modulation on the carrier. However, since the phase key $\phi_1, \phi_2, \ldots \phi_{(N-1)/2}$ is applied, the generated keyfunction has a kind of arbitrary shape. For each branch this keyfunction is time shifted with respect to the former branch by $\Delta t = 1/R_s$. This shift corresponds to the time from the maximum to the first zero crossing of

the former SPS. In the next step the lower bandwidth analog sub-signals are multiplied (X) with the time shifted keyfunctions and the result is added up ($\Sigma$) to build the arbitrary, high-bandwidth signal (**d**). This signal is then modulated on a carrier, which can be a wireless, THz or optical one and is transmitted (Tx) via the channel. On the receiver side (right), the received (Rx) and demodulated baseband signal is first power-split to the $N$ branches (1/$N$) and in each branch the whole signal is multiplied with the time-shifted keyfunctions. After analog to digital conversion (ADC) with a bandwidth of $B/N$, the sub-samples are restored and can be used to rebuild the original signal in the parallel-to-serial (P/S) converter (**e**). The splitter and combiner for conversion between signal and sub-signal are shown in yellow, and the application of phase key for en- and decryption are shown in blue.

## Experiments and simulations

In the proof-of-concept experiment, 12 Nyquist signals with a symbol rate of 22.5 GBd (and bandwidth of 22.5 GHz) were encrypted with a 15-line keyfunction and added up by a 12-line optical frequency comb with around 21 GHz spacing to a super-signal of 252.4 GHz bandwidth. The overall achieved symbol rate was accordingly $12 \cdot 22.5$ GBd = 270 GBd, corresponding to a 7% faster than Nyquist super-signal.

Please note that much higher data rates and bandwidths can be encrypted by simply increasing the number of comb lines. The rectangular bandwidth of the super-signal can be seen in Fig. 4a. Only if the keyfunction including bandwidth and center frequency of the signal are known, the signal can be decrypted, as shown in Fig. 4c for the measured bit error rate (BER) against the optical signal to noise ratio (OSNR). As an example, the measured eye diagram for an OSNR of 23.2 dB is shown in the inset (i) of Fig. 4c. A measured time trace when the key matching fails is also shown (ii) in Fig. 4c. If the keyfunction is known, but not the center frequency and bandwidth of the signal, a decryption is still not possible. Through simulations it is verified that the bandwidth of the signal has to be known with an accuracy of 6 MHz (Fig. 4b). The center frequency has to be known as well by a few MHz. However, if the keyfunction is known, the data can be detected at the receiver and the post processing software tries to adapt the local oscillator frequency to the carrier until a maximum eye opening is achieved. However, based on the experimental results, the required center frequency accuracy for the post-processing is within 100 MHz. Therefore, if the 252.4 GHz super-signal is divided into spectral slices for post-processing by filters, these filters must fit the bandwidth with an accuracy of 0.0024% and at the same time, that of the center frequency by 0.04%. Further details of

the simulations and theoretical analysis are provided in the supplementary material as Supplementary note 2 and Supplementary note 1, respectively.

En- and decryption was also performed in the baseband (Fig. 3c) for signals modulated with higher-order modulation formats. The spectrum of a 90 GBd 4-QAM super-signal encrypted with a 9-line keyfunction is shown in Fig. 4d. If the center frequency and bandwidth are known, but not the keyfunction, the signal decryption fails and cannot be detected, Fig. 4 (e). When the correct keyfunction, center frequency and bandwidth are known, the signal can be processed in 9 virtually-parallel digital branches. In each branch, the encrypted data is multiplied with time-shifted copies of the keyfunction, and the resulting constellation plots are shown in Fig. 4f.

## Conclusion

This paper introduces a method for the en- and decryption of signals based on the principles of signal theory. The signal is split into orthogonal sub-signals, the orthogonality of which is destroyed by a keyfunction for the transmission of a noise-like signal. On the receiver side, the signal is restored by the same keyfunction and a very precise filter. The encrypted signals do not need any additional bandwidth for transmission and the method can be straightforwardly implemented into existing optical or wireless communication systems by software or digital signal processing algorithms in the transmitter and receiver. For a key strength of $N_i = 201$, we get an estimated key space of around $10^{70}$.

The method can as well be used for the en- and decryption of very broadband faster than Nyquist super-signals. In this case the security of the encryption is additionally ensured by hiding the single signal in the spectrum of a noise-like super-signal much broader than the detection

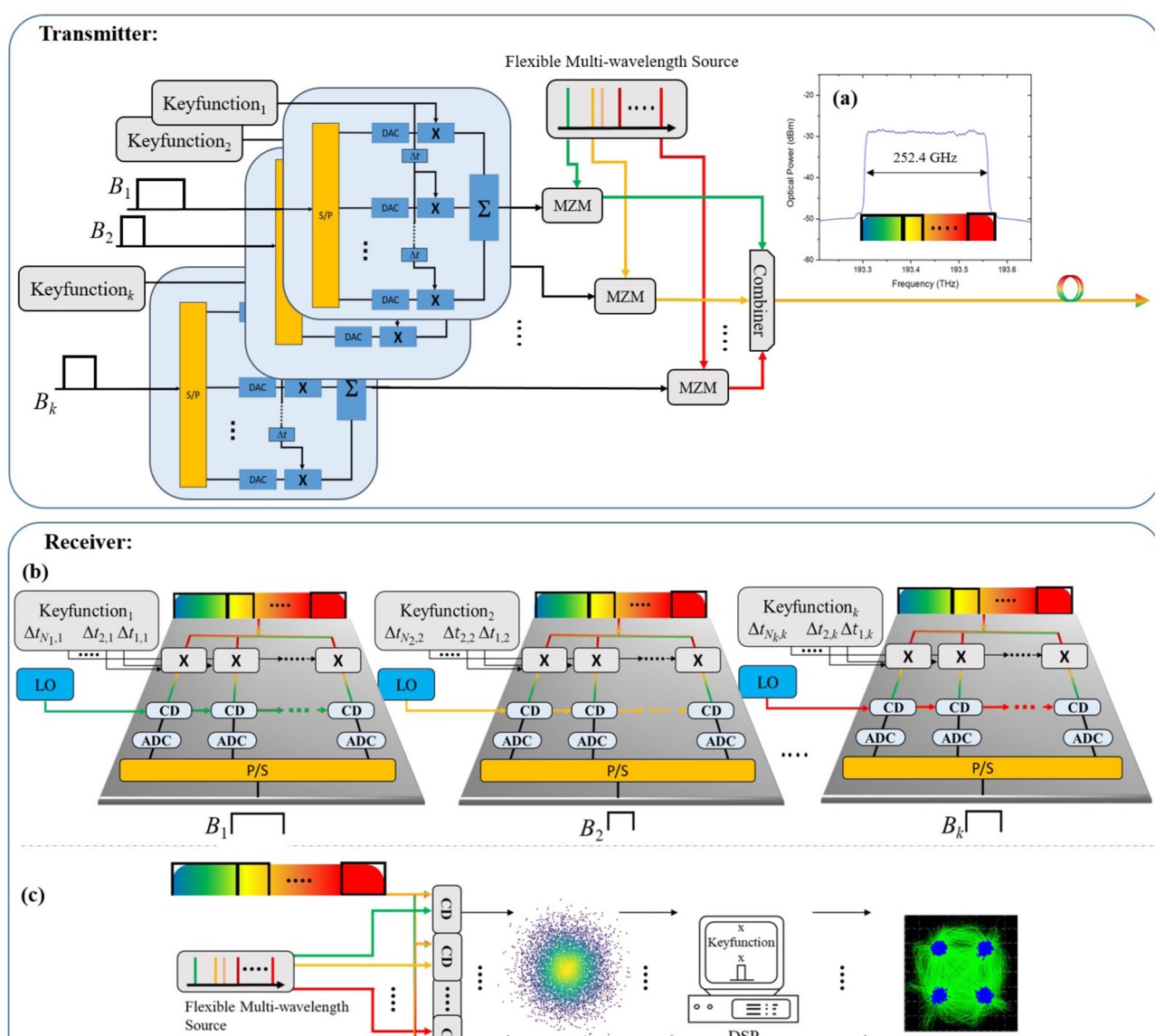

**Fig. 3 | High-bandwidth super-signal en- and decryption.** For the encryption of each single signal $B_1$, $B_2$, … $B_k$, the method described in Fig. 2 is utilized. The signals can be modulated using Mach-Zehnder modulators (MZMs) on single frequencies from a multi-wavelength source with unequal frequency spacing. The spacing between two frequencies in the source corresponds to the symbol rate of the single signal, so that a broadband rectangular super-signal without any guardband will be generated, an experimental example is shown in (**a**). On the receiver side, the whole broadband super-signal is multiplied with the respective keyfunction and detected with a coherent detector (CD) driven with a local oscillator (LO) wave close to the center wavelength of the respective signal in the rectangular spectrum (**b**). After analog-to-digital conversion in a low-bandwidth ADC and a subsequent digital filtering with the bandwidth of the sub-signal, the samples from the different branches are parallel-to-serial converted (P/S) and the original signal can be seen at the output. The signal can also be decrypted by applying the keyfunction in the baseband (**c**). Here again the LO wavelength must be close to the center frequency of the signal to be encrypted and can be generated by a similar multi-wavelength source. After coherent detection, the baseband signal is multiplied with time-shifted copies of the keyfunction, filtered and processed in parallel. This approach reduces the hardware complexity of the receiver and can realize encryption with key-functions with a large $N_i$. Please note that for both methods no optical filter is required to detect and decrypt the whole super-signal.

bandwidth of state of the art receivers and signal processing electronics. For detection, besides the keyfunction, the bandwidth and center frequency of the signal has to be known. For a bandwidth of 1 THz and a key strength of $N_i = 201$, the estimated key space is enhanced to around $10^{77}$. The presented method might offer a solution for secure data transmission in already existing and future optical, THz or wireless systems with low-latency and data rates in the terabits per second range.

## Methods

A schematic of the proof-of-concept encryption setup is shown in Fig. 5. To test the encryption for a densely packed broadband signal, a 12-line optical frequency comb with 20.9 GHz spacing was used. Four independent lasers, frequency separated by 62.7 GHz, were externally modulated by an MZM with a 20.9 GHz sine wave, to generate two sidebands around each laser line. To enable a flat optical comb, the bias of the MZM was adjusted in a way that the two generated sidebands have the same power as the laser lines. The encryption with a 15-line keyfunction was done by programming an arbitrary waveform generator, which drives the modulator. So, virtually, the 22.5 GBd data was divided into 15 parallel branches of 1.5 GBd (22.5 GBd/15) and each of these sub-signals is multiplied with time delayed copies of the 15-line keyfunction. Afterwards, they were added up and all 12 comb lines were modulated with the same encrypted 22.5 GBd binary phase shift keying

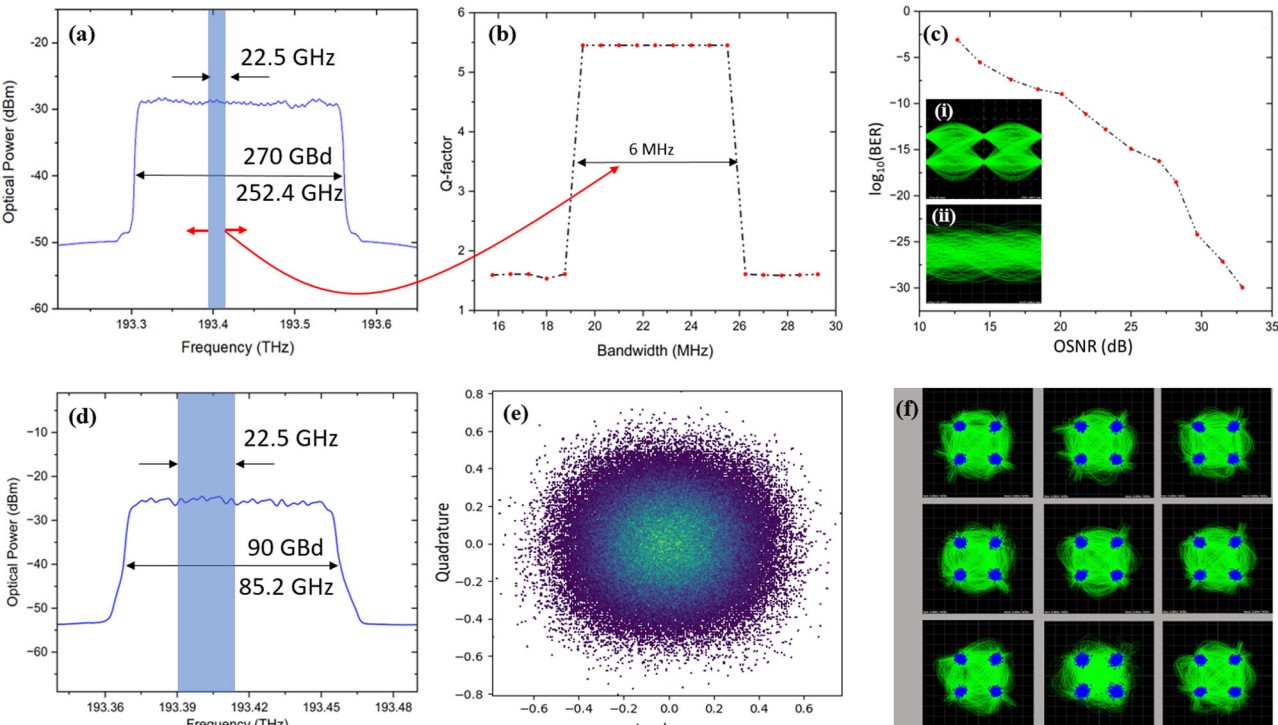

**Fig. 4 | Proof-of-concept results.** Experimental (**a**, **c**) and simulation results (**b**) for the encryption of a 252.4 GHz, 270 GBd faster than Nyquist super-signal consisting of 12 · 22.5 GBd rectangular signals with a keyfunction with a strength of $N_i$ = 15 lines. The measured bandwidth of the super-signal is presented in (**a**), and the single signal targeted for decryption is highlighted in blue (22.5 GBd). If all parameters are known (center frequency, bandwidth and keyfunction), the measured bit error rate (BER) against optical signal to noise ratio (OSNR) and the eye diagram for an OSNR of 23.2 dB are depicted in (**c**) and (i), respectively. The measured time trace for the BPSK signal when the keyfunction is not known is shown in (ii). The accuracy of the filter bandwidth required for spectrum slicing of the high-bandwidth signal (if the 15-line keyfunction and the center frequency are known) can be seen in (**b**). The measured signal spectrum for a 90 GBd QPSK data encrypted with $N_i$ = 9 lines is shown in (**d**) with the single signal targeted for decryption highlighted in blue (22.5 GBd). The decryption of the signal is carried out in the baseband as described in Fig. 3c. The simulated constellation diagram if the center frequency and bandwidth but not the keyfunction are known is presented in (**e**). The measured constellation diagrams, if all key-parameters are known, are shown in (**f**).

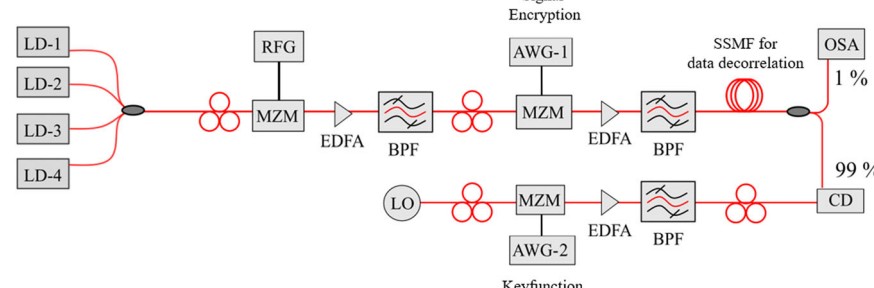

**Fig. 5 | Schematic of the experimental setup.** LD laser diode, MZM Mach Zehnder modulator, RFG radio frequency generator, AWG arbitrary waveform generator, BPF bandpass filter, EDFA erbium doped fiber amplifier, SSMF standard single mode fiber, LO local oscillator, CD coherent detector, DSP digital signal processing.

(BPSK) pseudo random bit sequence (PRBS-7). To de-correlate the data content of the 12 signals, a single mode fiber with a length of 33 km was used, which corresponds to a dispersion of 561 ps/nm.

With the setup in Fig. 5, only one sub-signal (22.5 GBd/15) can be detected at a given time. So, the whole 252.4 GHz signal is multiplied with the keyfunction, again generated by an AWG and modulated onto a local oscillator (LO) wave, in a coherent detector. The wavelength of the LO was adjusted very close to the center wavelength of the signal that has to be received. The whole 22.5 GBd signal can be decrypted by changing the time delay of the keyfunction in the AWG and the whole super-signal is detected by additionally changing the wavelength of the local oscillator. After detection, the signal is filtered in the baseband with a 1.5 GHz filter. The Q-factor performance against OSNR was characterized by adding a Gaussian white noise with different power levels. Please note that instead of multiplying the keyfunction with the LO wave, the whole signal can be multiplied with the keyfunction by a modulator in the signal path, as shown in Fig. 3.

For the second set of experiments, the decryption has been made by software algorithms in the baseband after down-conversion of the signal, as depicted in Fig. 3c. A 90 GBd quadrature phase shift keying (QPSK) PRBS-7 data was en- and decrypted using a 9-line keyfunction by this approach. The in-phase (I) and quadrature (Q) encrypted components were generated by programming the AWGs at a data rate of 22.5 GBd and modulated on the optical carriers by an IQ modulator. At the receiver, a tunable LO is adjusted to the central wavelength of the single signal in the super-signal that has to be decrypted. After coherent detection the signal is transferred to the digital domain by the analog to digital converter of a real-time oscilloscope. All following procedures are

baseband digital signal processing algorithms. So, the digital time trace is multiplied by the keyfunction with the right time shift and processed in nine parallel branches to get the 2.5 GBd (22.5 GBd/9) sub-signals. Since the multiplication with the keyfunction is realized by digital signal processing in the baseband, the parameters can be precisely tuned and locked to the detected signal. For the experiments, the resolution as well as clock and carrier recovery of the commercially available coherent detection system was sufficient. Transmitter and receiver were not synchronized.

The methods employed for the en- and decryption were realized by a commercially available communication testbed from Tektronix (OM4245, OM5110 and DPO73304). Forward error correction, pre-distortion or dispersion compensation were not employed in the experiments.

## Data availability
The datasets generated during and/or analyzed during the current study are available from the corresponding author on reasonable request.

## Code availability
The codes that were used for this study are available from the corresponding author upon reasonable request.

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

## Acknowledgements
This work was supported by the Deutsche Forschungsgemeinschaft (DFG, German Research Foundation) (454954953, 403154102, 491066027).

## Author contributions
T.S. proposed and supervised the project. A.V., K.S., and J.M. designed the system and conducted the experiments. A.V. and K.S. did the simulation and data analysis. J.M. developed the theoretical description. All authors contributed to the discussion of the results and the writing of the manuscript.

## Funding

## Competing interests

The authors declare no competing interests.
