## [Peer review file · Communications Engineering]

Signal Theory Based Encryption of Faster-than-Nyquist Signals for Fiber and Wireless Transmission

Corresponding Author: Mr Abhinand Venugopalan

Version 0:

Reviewer comments:

Reviewer #2

(Remarks to the Author)

Overview

This manuscript presents a proof-of-concept for encrypting Faster-than-Nyquist (FTN) super-signals using optical frequency combs and dynamic key functions. The authors demonstrate the creation and encryption of a high-bandwidth super-signal (exceeding the Nyquist limit by 7%, achieving a symbol rate of 270 GBd with a total bandwidth of 252.4 GHz). They also introduce an encryption method that not only scrambles the signal using dynamic key functions but also utilizes multiple frequency lines generated by an optical comb to modulate the signal. The key contribution of the paper is the multi-layered encryption approach that requires precise knowledge of key functions, bandwidth, and center frequency for decryption.

The concept of encrypting FTN super-signals using optical frequency combs and dynamic keyfunctions is novel and highly relevant to the field of optical communications. The manuscript can be considered innovative because it combines FTN transmission with a dynamic encryption mechanism that is hardware-driven and provides a robust solution for protecting high-speed communication.

The major claims for the manuscript can be summarized as follows

-The authors claim to have successfully encrypted an FTN super-signal with a bandwidth of 252.4 GHz and a symbol rate of 270 GBd. They highlight that this is the first time such a high-bandwidth FTN super-signal has been fully encrypted using their method.

-The paper asserts that even if attackers know the key function, they cannot decrypt the signal unless they also know the precise bandwidth (within 6 MHz accuracy) and center frequency (within a few MHz accuracy).

-The authors suggest that their method is highly scalable and that higher data rates and bandwidths can be achieved by simply increasing the number of comb lines in the optical frequency comb.

Overall, the manuscript presents an advancement in the field of encrypted high-bandwidth communications, particularly through its application of FTN super-signal encryption. However, there are some limitations regarding practical implementation and the need for further experimentation with higher-order modulations that should be carefully addressed before possible publication. Please refer to the comments below when revising the manuscript.

1- The proposed method assumes that once orthogonality is destroyed, the signal becomes noise-like and completely undetectable. However, this overlooks the possibility of advancements in noise reduction or signal interpretation techniques that might detect underlying structures even in noise.

2- On the receiver side, the inverse process of combining and decrypting the sub-signals may introduce complexity. Small errors in the key functions, time shifts, or filter mismatches could potentially lead to signal distortion or loss of data. Moreover, the spacing between the frequency lines generated by the comb must be very precise, because any misalignment or noise in the frequency generation process could disrupt the accurate modulation.

3- Regarding the previous comment, splitting signals into branches (each requiring precise keyfunction multiplication and convolution) would add significant complexity to the decryption process. The authors are recommended to study this

complexity and to clearly discuss about its effect on the system.

4- Since the security of the entire encryption system relies on the secure channel used for transmitting key information, how is this channel protected against potential attacks?

5- The need for extreme precision in both bandwidth and center frequency might make the system vulnerable to environmental factors, such as signal drift, noise, or imperfections in hardware components. If the receiver's equipment can't maintain this level of accuracy, the system's reliability might be compromised.

6 - Given the widespread use of OFDM in modern optical communication systems, how well would the proposed encryption method integrate with OFDM modulation? Would any modifications be necessary to apply the keyfunction-based encryption approach to OFDM's orthogonal sub-carriers?

Reviewer #3

(Remarks to the Author)

The authors have proposed and demonstrated a novel encryption method based on signal theory for faster than Nyquist superchannels. The method is based on two fundamental ideas; 1) create a super-signal by combining multiple rectangular-bandwidth Nyquist signals so that the overall signal bandwidth exceeds the bandwidth of typical receivers and storage devices making it difficult to detect, store and post process 2) to break/destroy the orthogonality of sampling points rendering transmitted signal look-like noise due to interference between sampling points and restoring orthogonality at the receiver through identical predetermined (mutually coordinated) keyfunctions. The authors claim the proposed method is agile, transparent to any modulation format, does not require additional bandwidth and is easy to implement in any communication system through hardware or software processing.

Overall, the manuscript is well organized, easy to read and formatted except some figures. The proposed method is novel and comparable to existing methods for encryption and for sure will be of interest for the relevant scientific community. Most of claims are supported by mathematical description, simulations and experiments results.

The manuscript can be accepted if following comments are addressed.

I have the following comments for the authors:

1. The authors point out that the only possibility for attack is through spectrum slicing and it seems the one of the underlying assumptions is that spectrum slicing is difficult (I may not have understood correctly). There has been a recent paper (ECOC24) where the authors demonstrated spectrally-slice coherent receiver with a record bandwidth of 2.4 THz. How will it affect the strength of the proposed method?
2. Since rectangular-bandwidth Nyquist signals can have unequal bandwidths (size), carriers will also have unequal spacing. The authors have proposed to use frequency combs which inherently exhibit equal carrier spacing. It would be more appropriate to use terms like "flexible carrier source" or "flexible multiwavelength source" that can be built either by using combination of a frequency comb and flexible multiwavelength filter (e.g., Wavelength selected switches (WSS)) OR by an array of integrated tunable lasers OR any other implementation providing asymmetric carrier spacing and central frequency tunability.
3. Figure 3 must be improved/redrawn, it is not easy to understand for the reader. To start with, I would suggest differentiating optical and electrical links/connections with different colours or line style. For example, in transmitter, electrical inputs and optical outputs of MZMs are shown in same colour. Same is the case for concatenation/summation of multiple Nyquist signals. The part describing the receiver must be better elaborated, may be using sub-figures if authors prefer to keep block level representation.
4. The authors have proposed two methods for decryption process: 1) multiplying key functions with the copies of whole spectrum 2) multiplying key functions in baseband post detection and down-conversion. The second approach is also interesting and probably more practical therefore should also be depicted in Figure 3. In addition, the explanation can also be improved.
5. How does the proposed en- decryption method affect the overall transmission system complexity in terms of additional hardware, optoelectronic component's quality (precision) and computation complexity. For example, laser drift, roll-off factor of optical filters, need of additional optical modulators for key-function multiplication in the receiver etc.
6. Line 190? the authors described that keyfunctions can be changed by phase modulation. Is it trivial to do in real time/ in-service?
7. Line 92: The agility in terms of keyfunctions, bandwidth and central frequency is constrained by the need of a secure channel. Can authors comment on it? How it compares with other techniques like steganography, spread spectrum and QKD etc?
8. In case of low bandwidth signals, authors suggest filling the rectangular spectrum with noise or fake signals. Wouldn't it make transmission system spectrally inefficient by reducing the overall spectral efficiency? What about using flexible carrier source and baud rates to adjust the bandwidth according to signal bandwidth?
9. What about using a shared/ identical comb source (flexible carrier source) like the one in trasmitter as an LO in the receiver for CD?

Reviewer #4

(Remarks to the Author)

In this manuscript, authors claim that they have developed a signal theory based encryption method in the physical layer to ensure security of fiber and wireless communication systems.

After carefully reading the manuscript, I think the manuscript is not suitable for publication.

First of all, the motivation of this work is doubtful. To ensure the security of data, current communication systems implement digital data encryption at higher layer of OSI stack -- for a good reason. Not all communication data needs to be encrypted, and not all data that needs encryption requires the same level of security and encryption. When high level of security is needed, more sophisticated encryption scheme can be used (and is available) against hostile attack. It would be a huge waste of resource to implement encryption or other security measures at physical layer for all data transmitted, which is what this manuscript suggests.

Secondly, the use of the term encryption in this manuscript is unorthodox. We normally refer to encryption as the digital data encryption. The idea suggested by the authors is what we normally call physical layer security, but we rarely refer to it as encryption. Physical layer security methods have been proposed for very specific applications such as in electronic warfare scenarios, they are not suitable for other or more general applications.

Finally and most importantly, the authors fail to clearly describe their method and the uniqueness and advantages of their scheme over existing physical layer security schemes. The description given in the first paragraph of the section "Principle of en- and decryption" is very vague. The claim of a 252.4 GHz super-signal with a 7% higher symbol rate than given by the Nyquist limit is simply not possible -- did the authors find a way to break the Nyquist sampling theorem? From what I can gather in the description of the proposed method, it is no different than applying a frequency domain scrambler with a key function, which is nothing new and has been proposed by other works already.

In summary, I don't recommend publication of this manuscript in its current form.

Version 1:

Reviewer comments:

Reviewer #2

(Remarks to the Author)

The reviewer thanks the authors for their effort and for preparing the revision. My comments and concerns in the previous round have been adequately addressed and I have no further comments.

Reviewer #3

(Remarks to the Author)

The authors have made necessary revisions in the manuscript and provided satisfactory reply/explanation to suggestions and queries.

The manuscript can be accepted for publication.

Reviewer #4

(Remarks to the Author)

I am satisfied with the authors' modification of the paper according to reviewers' comments. Thanks to the authors' effort to clear some definitions and the overall organization of the paper. I think the manuscript is now ready for publication.

Authors Response Letter (Manuscript ID: COMMS-24-0450-T)

Dear editor and reviewers,

Thank you for your decision letter dated 01.11.2024 and giving us the opportunity to revise and resubmit our research article entitled “Signal Theory Based Encryption of Faster-than-Nyquist Super-Signals for the Next-Generation Fiber and Wireless Infrastructure” (Manuscript ID: COMMS-24-0450-T). As response of a comment from one of the reviewers we have changed the title to: “Signal Theory Based Encryption of Faster-than-Nyquist Super-Signals for Next-Generation Fiber and Wireless Systems”. We would like to thank the reviewers for their efforts and careful reviews of this paper. The suggestions offered by the reviewers have been immensely helpful.

We have included the reviewers’ comments immediately after and responded to them individually, indicating exactly how we addressed each concern or problem and describing the changes we made. For your convenience, in this response letter we have included the reviewer’s original comments in *italic*, with our answer in **bold**, and the corresponding changes in the manuscript are indicated in **blue**.

-----Reviewer Comments-----

Reviewer 2:

Overview

This manuscript presents a proof-of-concept for encrypting Faster-than-Nyquist (FTN) super-signals using optical frequency combs and dynamic key functions. The authors demonstrate the creation and encryption of a high-bandwidth super-signal (exceeding the Nyquist limit by 7%, achieving a symbol rate of 270 GBd with a total bandwidth of 252.4 GHz). They also introduce an encryption method that not only scrambles the signal using dynamic key functions but also utilizes multiple frequency lines generated by an optical comb to modulate the signal. The key contribution of the paper is the multi-layered encryption approach that requires precise knowledge of key functions, bandwidth, and center frequency for decryption.

The concept of encrypting FTN super-signals using optical frequency combs and dynamic keyfunctions is novel and highly relevant to the field of optical communications. The manuscript can be considered innovative because it combines FTN transmission with a dynamic encryption mechanism that is hardware-driven and provides a robust solution for protecting high-speed communication.

The major claims for the manuscript can be summarized as follows

-The authors claim to have successfully encrypted an FTN super-signal with a bandwidth of 252.4 GHz and a symbol rate of 270 GBd. They highlight that this is the first time such a high-bandwidth FTN super-signal has been fully encrypted using their method.

-The paper asserts that even if attackers know the key function, they cannot decrypt the signal unless they also know the precise bandwidth (within 6 MHz accuracy) and center frequency (within a few MHz accuracy).

-The authors suggest that their method is highly scalable and that higher data rates and bandwidths can be achieved by simply increasing the number of comb lines in the optical frequency comb.

Overall, the manuscript presents an advancement in the field of encrypted high-bandwidth communications, particularly through its application of FTN super-signal encryption. However, there are some limitations regarding practical implementation and the need for further experimentation with higher-order modulations that should be carefully addressed before possible publication. Please refer to the comments below when revising the manuscript.

We thank the reviewer for the insightful comments. To show that the method can be quite easily implemented in practical systems, we have included new experimental results with a decryption of the transmitted data signal directly in the baseband in the revised version. If only a single channel has to be encrypted, this can be implemented by a software update. For the encryption of super-signals, a frequency comb or flexible multi-wavelength source is necessary at the transmitter and receiver. However, for the baseband decryption no branches, modulators or other hardware is necessary. Additionally, no AWG is needed on the receiver side. Therefore, we were able to show the encryption of signals with higher order modulation format in the new version of the paper.

1- The proposed method assumes that once orthogonality is destroyed, the signal becomes noise-like and completely undetectable. However, this overlooks the possibility of advancements in noise reduction or signal interpretation techniques that might detect underlying structures even in noise.

We thank the reviewer for this comment. For the mitigation of the noise in weak signals an active amplification, signal averaging and bandpass filtering can be used. However, these techniques cannot be used for the recovering of a non-repetitive waveform which is completely below the noise level [1]. Recently, the Talbot effect has been used to detect a signal buried under in-band noise, and for sensing applications squeezed states of light can be incorporated to increase the signal to noise ratio [1, 2]. However, all these methods are based on the fact that the signal is still there but buried under noise. In our method there are no underlying signal or other structures anymore. The signal itself is transformed into noise by destroying the orthogonality between the sinc-pulse sequences.

To make that more clear, we have included a similar discussion into the “Principle of en- and decryption” section:

Figure 1 (d) shows the signal after the same keyfunction has been applied to the frequencies of the nine-line SPSs. Due to the keyfunction, the phase of the frequency lines is not equal anymore, the orthogonality is destroyed and, due to intersymbol interference, the superposition is a kind of arbitrary, noise-like signal. Since no underlying structures are present, methods to recover a signal buried under noise cannot be used ²⁹.

2- On the receiver side, the inverse process of combining and decrypting the sub-signals may introduce complexity. Small errors in the key functions, time shifts, or filter mismatches could potentially lead to signal distortion or loss of data. Moreover, the spacing between the frequency lines generated by the comb must be very precise, because any misalignment or noise in the frequency generation process could disrupt the accurate modulation.

Please see also answer 5 for reviewer #3

We agree with the reviewer that the security of the method depends on the matching of the post-processing filter, the local oscillator, the key function and time shift in the receiver. Therefore, all these parameters have to be aligned. However, in our experiments the resolution of the commercially available coherent detector system (Tektronix OM4245 Optical Modulation Analyzer) was sufficient for a successful decryption of the signal. But, for the sake of experimental simplicity in the first proof-of-concept experiments we have synchronized the key function generator in the transmitter and receiver to have the correct time shift. In our new experiments, presented in the new version of the paper however, transmitter and receiver were completely independent and we are using a post-processing in the baseband for the decryption of the signals. In this case it is sufficient to adjust all known parameters and scan the time-shift in the receiver until a signal can be detected. After that, the clock recovery in the receiver fulfills the synchronization.

In the presented method we have used two combs. One for the key function and the other one for generating a high-bandwidth faster than Nyquist super-signal. The comb for the key function is generated by software or digital signal processing in an arbitrary waveform generator (or for our new set of experiments by digital signal processing in the receiver) and is therefore not prone to any physical changes. The alignment of the frequency lines in the high-bandwidth comb for the super-signal and for the local oscillator is not that important. Once the commercial detector is adjusted to decrypt and receive the signal, its internal signal processing for carrier recovery is able to follow changes of the carrier frequency.

Additionally, since we are using a faster-than-Nyquist transmission, we have an overlap between the sub-signals anyway. Up to a certain extent [3] a change of this overlap results in a change of the bit error rate, which can be compensated by a forward error correction. However, when the spacing between the frequency lines becomes too small or too wide, the bandwidth of the signals can be detected in the spectrum of the super-signal, which reduces the security of the method. But, if the comb is generated by an electro-optic modulation, the frequency spacing is defined by the RF signal, which can be adjusted very precisely and even for integrated comb sources based on ring resonators for instance, a very precise adjustment and very high quality of the comb lines is possible [4].

In the new version of the paper, we have added the following discussion in the Methods section:

Please note that instead of multiplying the keyfunction with the LO wave, the whole signal can be multiplied with the keyfunction by a modulator in the signal path, as shown in Fig. 3.

For the second set of experiments, the decryption has been made by software algorithms in the baseband after down-conversion of the signal, as depicted in Fig. 3 (c). A 90 GBd quadrature phase shift keying (QPSK) PRBS-7 data was en- and decrypted using a 9-line keyfunction by this approach. The in-phase (I) and quadrature (Q) encrypted components were generated by programming the AWGs at a data rate of 22.5 GBd and modulated on the optical carriers by an IQ modulator. At the receiver, a tunable LO is adjusted to the central wavelength of the single signal in the super-signal that has to be decrypted. After coherent detection the signal is transferred to the digital domain by the analog to digital converter of a real-time oscilloscope. All following procedures are baseband digital signal processing algorithms. So, the digital time trace is multiplied by the keyfunction with the right time shift and processed in nine parallel branches to get the 2.5 GBd (22.5 GBd/9) sub-signals. Since the multiplication with the keyfunction is realized by digital signal processing in the baseband, the parameters can be precisely tuned and locked to the detected signal. For the experiments, the resolution and clock recovery of the commercially available coherent detection system was sufficient. Transmitter and receiver were not synchronized.

The methods employed for the en- and decryption were realized by a commercially available communication testbed from Tektronix (OM4245, OM5110 and DPO73304). Forward error correction, pre-distortion or dispersion compensation were not employed in the experiments.

3- Regarding the previous comment, splitting signals into branches (each requiring precise keyfunction multiplication and convolution) would add significant complexity to the decryption process. The authors are recommended to study this complexity and to clearly discuss about its effect on the system.

We completely agree with the reviewer that a single branch for each line in the keyfunction would either reduce the security of the method, since only a few branches are realized, or the complexity for a high number of branches. Additionally, the security might be reduced as well if the number of branches and therefore lines in the keyfunction is fixed and defined by hardware. Therefore, in our new experiments, which we have included in the new version of the paper, we have used a digital processing of the baseband signal in the receiver to decrypt the signal. The baseband signal is multiplied with the keyfunction by an algorithm directly in the real-time oscilloscope. After this first step the other common signal processing algorithms for clock and carrier recovery, resampling and so on are carried out. This slightly increases the complexity of the electronic signal processing by a multiplication and the processing in virtual or real parallel electronic branches. However, even if N real parallel electronic branches are incorporated, the sampling rate and bandwidth for each branch is reduced by the same number N . So this higher complexity in parallel electronic branches is at least partly compensated by lower sampling rate and bandwidth in each single branch.

The super-signal consists of k signals. So, the bandwidth of the super-signal is $k \times B$. But, for the decryption of the single signal in the baseband no additional branches are needed. It is sufficient to adjust the local oscillator wave from a comb or a flexible carrier source to the signal to be encrypted. If the bandwidth and keyfunction are known, the carrier recovery in the receiver is able to follow frequency changes of the carrier. However, for very high-bandwidth signals (B), the digital signal processing in virtual-parallel branches and the bandwidth reduction for the detection and processing can be further enhanced by additional parallel hardware branches. So for a keyfunction with $M \times N$ frequency lines, a number of M hardware branches with an optical modulator for the multiplication of the received signal with an M -line keyfunction reduces the bandwidth for the following detection and digital signal processing

by B/M . For decrypting this signal, the N line keyfunction can then be applied in the software. This number N is not fixed and can simply be changed by the software.

We have further explained this in the new version of the paper:

In the receivers for each of the sub-signals (Fig. 3(b)) the whole spectrum will be detected at once and power-split into a number of branches which correspond to the key strength N_i . In each single branch the whole spectrum is multiplied with the respective keyfunction. In the first set of experiments, this multiplication has been done with MZM, where the optical input is the whole signal spectrum and the electrical input is the keyfunction. This multiplication in parallel branches reduces the hardware complexity at the receiver. Since the sub-signal bandwidth is N_i -times lower than the signal bandwidth and kN_i times lower than that of the super-signal, the signal to noise and distortion ratio and effective number of bit of the ADC is increased²⁸ and can in turn at least partly compensate for the increase in power consumption of the parallel signal processing. However, this means that the number of branches N_i and therefore the key strength is fixed and defined by the hardware. Alternatively, the multiplication can be done in the baseband after detection and down-conversion, as done in the second series of experiments and described in Fig. 3 (c). In the case of multiplication in the baseband, the number N_i and therefore the key strength can be very high and it can be varied, even during the transmission. A mixture between both approaches is possible as well. So for a keyfunction with $M \times N_i$ frequency lines, a number of M hardware branches with an optical modulator for the multiplication of the received signal with an M -line keyfunction reduces the bandwidth by M for the following detection and electronic processing. For decrypting this signal, the additional N_i -line keyfunction can be applied in the software. This number N_i is not fixed and can be changed.

4- Since the security of the entire encryption system relies on the secure channel used for transmitting key information, how is this channel protected against potential attacks?

Please see also answer 7, reviewer #3.

If the secure channel is compromised, the encrypted signal can be decrypted by an ineligible user. Therefore, it has to be very safe. However, the data rate and therefore bandwidth of that secure channel can be quite small. Therefore, it can be a quantum key distribution [6], for instance. If the low-bandwidth secure channel exists for the time of the transmission in the high-bandwidth channel, all parameters of the key can be changed during transmission. Another possibility for permanently changing the encryption is to use a pre-defined open signal which is available and equal at the transmitter and receiver side. This can be a broadcast radio or TV signal, any signal from any available network, or noise from a source which can be detected and is equal at both sides.

If the secure channel does not exist for the whole time, besides quantum key distribution, electronic key distribution methods like post-quantum cryptographic standards [7] or even standard public-private key distribution may be used. However, for the latter it means that the whole encryption is only as safe as the key distribution. The following is included in the revised manuscript:

However, at every time the keyfunction including bandwidth, as well as center frequency of the signal must be known by the receiver. Therefore, a secure channel is required for the transmission of this information. Since the parameters for the encryption have a low bandwidth and the secure channel has to exist for a short time only, secure information exchange via QKD, post quantum cryptography or even hardware security primitives can be considered for this.

5- The need for extreme precision in both bandwidth and center frequency might make the system vulnerable to environmental factors, such as signal drift, noise, or imperfections in hardware components. If the receiver's equipment can't maintain this level of accuracy, the system's reliability might be compromised.

In our new experiments we have used a completely free-running receiver without any synchronization to the transmitter. Once the parameters for decryption are adjusted and the receiver detects a signal, the signal processing in the receiver is able to follow any changes. The receiver is part of a commercial communications testbed from Tektronix (OM4245, OM5110 and DPO73304) and able to decrypt the signals without error over a long time span. In the new version of the paper we have included:

Since the multiplication with the keyfunction is realized by digital signal processing in the baseband, the parameters can be precisely tuned and locked to the detected signal. For the experiments, the resolution as well as clock and carrier recovery of the commercially available coherent detection system was sufficient. Transmitter and receiver were not synchronized.

The methods employed for the en- and decryption were realized by a commercially available communication testbed from Tektronix (OM4245, OM5110 and DPO73304). Forward error correction, pre-distortion or dispersion compensation were not employed in the experiments.

6 - Given the widespread use of OFDM in modern optical communication systems, how well would the proposed encryption method integrate with OFDM modulation? Would any modifications be necessary to apply the keyfunction-based encryption approach to OFDM's orthogonal sub-carriers?

We thank the reviewer for the interesting question. The method is completely transparent for the signal. So, any kind of signal, including OFDM can be en- and decrypted without any modifications. As any signal, an OFDM signal is a time trace which can be sampled error-free as long as the sampling theorem is not violated. The multiplication with the key function can be seen as such a sampling process. We have not used the encryption method with OFDM signals yet. However, in [8] we have shown the sampling and multiplexing of OFDM signals with non-encrypted sinc-pulse sequences.

In the paper we have included the following discussion:

Here, a real-time encryption method based on the fundamentals of signal theory is proposed. The encrypted signal does not need any additional bandwidth, and the encryption can be easily implemented in any communication system, either through hardware or software processing, suitable for the next-generation fiber and wireless infrastructure. The encryption method is completely transparent to any modulation format and even orthogonal frequency division multiplexed (OFDM), analog radio over fiber signals, or signals already encrypted at a higher OSI level, can be encrypted.

Reviewer #3 (Remarks to the Author):

The authors have proposed and demonstrated a novel encryption method based on signal theory for faster than Nyquist superchannels. The method is based on two fundamental ideas; 1) create a super-signal by combining multiple rectangular-bandwidth Nyquist signals so that the overall signal bandwidth exceeds the bandwidth of typical receivers and storage devices making it difficult to detect, store and post process 2) to break/destroy the orthogonality of sampling points rendering transmitted signal look-like noise due to interference between sampling points and restoring orthogonality at the receiver through identical predetermined (mutually coordinated) keyfunctions. The authors claim the proposed method is agile, transparent to any modulation format, does not require additional bandwidth and is easy to implement in any communication system through hardware or software processing.

Overall, the manuscript is well organized, easy to read and formatted except some figures. The proposed method is novel and comparable to existing methods for encryption and for sure will be of interest for the relevant scientific community. Most of claims are supported by mathematical description, simulations and experiments results.

We thank the reviewer for these comments.

The manuscript can be accepted if following comments are addressed.

I have the following comments for the authors:

1. The authors point out that the only possibility for attack is through spectrum slicing and it seems the one of the underlying assumptions is that spectrum slicing is difficult (I may not have understood correctly). There has been a recent paper (ECOC24) where the authors demonstrated spectrally-slice coherent receiver with a record bandwidth of 2.4 THz. How will it affect the strength of the proposed method?

We thank the reviewer for this comment, and completely agree that we have to include that in our discussion. Indeed, the basic idea of spectrum slicing is to get the whole information for the broadband signal by a massive post processing of the recorded spectral slices. The spectral slices are overlapping and can be virtually stitched together by a computer. Therefore, if the complete information over the whole bandwidth is available, a massive computational power might enable to find the right bandwidth and center frequency. However, this can only be done by trying the different key possibilities for all possible

center frequencies and bandwidths with a resolution of 6 MHz. For example, the number of variations which have to be tried for a 1 THz signal and $N = 201$ is estimated to be of the order 10^{77} .

We have included this discussion in the new version of the paper:

The only possibility for an attack is to divide the real time, high-bandwidth super-signal into spectral slices by filters and electronic signal processing²⁴. However, as we will show, to successfully decrypt the signal, or the signals, the bandwidth of these filters can differ by only 10^{-3} % to that of the encrypted signal. Moreover, even when the whole signal spectrum is reconstructed by spectral slicing, a decryption might only be possible by trying the different key possibilities for all possible center frequencies and bandwidths within the encrypted super-signal.

2. Since rectangular-bandwidth Nyquist signals can have unequal bandwidths (size), carriers will also have unequal spacing. The authors have proposed to use frequency combs which inherently exhibit equal carrier spacing. It would be more appropriate to use terms like “flexible carrier source” or “flexible multiwavelength source” that can be built either by using combination of a frequency comb and flexible multiwavelength filter (e.g., Wavelength selected switches (WSS)) OR by an array of integrated tunable lasers OR any other implementation providing asymmetric carrier spacing and central frequency tunability.

We thank the reviewer for this suggestion and have included this in the new version of the paper:

With an additional comb source with k frequency lines or a flexible multi-wavelength source this basic idea can be used to generate very broadband encrypted super-signals with the overall bandwidth $B_{SE} = \sum_{i=1}^k B_i$, as shown for optical signals in Fig. 3. For reaching the maximum spectral efficiency, each signal can be chosen to be a Nyquist data signal, so that the bandwidth shape of each of the signals is rectangular and corresponds to the symbol rate. Ideally, the symbol rates and bandwidths are different and the frequency spacing between any two adjacent lines of the flexible multi-wavelength source should be adapted to the different symbol rates, so that the spacing between the signals is unequal and no guardband between the signal spectra is measurable in the super-signal.

In the text and figs. we have used the terms “flexible carrier source” or “flexible multi-wavelength source”.

3. Figure 3 must be improved/ redrawn, it is not easy to understand for the reader. To start with, I would suggest differentiating optical and electrical links/connections with different colours or line style. For example, in transmitter, electrical inputs and optical outputs of MZMs are shown in same colour. Same is the case for concatenation/summation of multiple Nyquist signals. The part describing the receiver must be better elaborated, may be using sub-figures if authors prefer to keep block level representation.

4. The authors have proposed two methods for decryption process: 1) multiplying key functions with the copies of whole spectrum 2) multiplying key functions in baseband post detection and down-conversion. The second approach is also interesting and probably more practical therefore should also be depicted in Figure 3. In addition, the explanation can also be improved.

We have redrawn the figure according to the suggestions and we have improved the explanation in the caption. Additionally, we have included the baseband encryption in Fig. 3 (c) and we describe it in the caption and the text. Due to the baseband encryption, an arbitrary waveform generator (AWG) is no longer required at the receiver side. Therefore, we can use our second AWG for the generation of 4-QAM signals.

Fig. 3 High-bandwidth super-signal encryption. For the encryption of each single signal B_1, B_2, \dots, B_k , the method described in Fig. 2 is utilized. Since Nyquist data signals were used before encryption, each signal has a rectangular and ideally a different bandwidth. The signals can be modulated on single frequencies from a multi-wavelength source with unequal frequency spacing. The spacing between two frequencies in the source corresponds to the symbol rate of the single signal, so that a broadband rectangular super-signal without any guardband will be generated, an experimental example is shown in (a). On the receiver side the whole broadband super-signal is multiplied with the respective keyfunction and detected with a coherent detector (CD) driven with a local oscillator (LO) wave close to the center wavelength of the respective signal in the rectangular spectrum (b). After analog-to-digital conversion in a low-bandwidth ADC and a subsequent digital filtering with the bandwidth of the sub-signal, the samples from the different branches are parallel-to-serial converted (P/S) and the original signal can be seen at the output. The signal can also be decrypted by applying the keyfunction in the baseband (c). Here again the LO wavelength must be close to the center frequency of the signal to be encrypted and can be generated by a multi-wavelength source. After coherent detection, the baseband signal is multiplied with time-shifted copies of the keyfunction, filtered and processed in parallel. This approach reduces the hardware complexity of the receiver and can realize encryption with keyfunctions with a large N . Please note that for both methods no optical filter is required to detect and decrypt the whole super-signal.

Fig. 4 Experimental (a, c) and simulation results (b) for the encryption of a 252.4 GHz, 270 GBd faster than Nyquist super-signal consisting of $12 \cdot 22.5$ GBd rectangular signals with a keyfunction with a strength of $N_k = 15$ lines. The measured bandwidth of the super-signal is presented in (a), and the single signal targeted for decryption is highlighted in blue (22.5 GBd). If all parameters are known (center frequency, bandwidth and keyfunction), the measured bit error rate (BER) against optical signal to noise ratio (OSNR) and the eye diagram for an OSNR of 23.2 dB are depicted in (c) and (i), respectively. The measured time trace for the BPSK signal when the keyfunction is not known is shown in (ii). The accuracy of the filter bandwidth required for spectrum slicing of the high-bandwidth signal (if the 15 line keyfunction and the center frequency are known) can be seen in (b). The measured signal spectrum for a 90 GBd QPSK data encrypted with $N_k = 9$ lines is shown in (d) with the single signal targeted for decryption highlighted in blue (22.5 GBd). The decryption of the signal is carried out in the baseband as described in Fig.3 (c). The simulated constellation diagram if the center frequency and bandwidth but not the keyfunction are known is presented in (e). The measured constellation diagrams if all key-parameters are known are shown in (f).

5. How does the proposed en- decryption method affect the overall transmission system complexity in terms of additional hardware, optoelectronic component's quality (precision) and computation complexity. For example, laser drift, roll-off factor of optical filters, need of additional optical modulators for key-function multiplication in the receiver etc.

Please see also answer 5 for reviewer #2

There are different possibilities to adapt the method into a transmission system and they require different complexity. If only a single standard 50 GHz channel, for instance, is en- and decrypted no additional hardware is needed. The en- and decryption can be fully done in the baseband by a software update of the transmitter and receiver.

If several sub-channels with equal or different bandwidth are stitched together, or if some encrypted sub-channels are transmitted in a broader rectangular bandwidth with additional noise, these sub-channels have to be stitched together in the transmitter. Therefore, a source that delivers several carrier frequencies is required. The frequency separation between these carriers should be stabilized to a certain degree. An overlapping of the channels increases the bit error rate, but this can be compensated by a forward error correction [3]. But, if the channels are too close together or too far away from each other, their bandwidth will be revealed by interference peaks or gaps in the spectrum.

On the receiver side another multi carrier source is needed to receive the whole super-signal. If just a single signal from the super-signal has to be received, it can be decrypted from the broadband spectrum by adjusting a local oscillator. The multiplication with the keyfunction and the filtering is carried out in the baseband. Recently we have shown the reception of a non-encrypted 662.5 GHz signal without any hardware filter [9].

For the experiments we have used a commercial communication testbed from Tektronix. For the newly presented results the transmitter and receiver were completely independent of each other. If the

decryption parameters are adjusted in the receiver, the carrier and clock recovery of the commercial system are sufficient to receive the signals error-free for hours. Since our equipment is already ten years old, we believe that today's commercial field equipment shows at least a similar performance.

The following is included in the paper:

Since the multiplication with the keyfunction is realized by digital signal processing in the baseband, the parameters can be precisely tuned and locked to the detected signal. For the experiments, the resolution as well as the carrier and clock recovery of the commercially available coherent detection system was sufficient. Transmitter and receiver were not synchronized.

The methods employed for the en- and decryption were realized by a commercially available communication testbed from Tektronix (OM4245, OM5110 and DPO73304). Forward error correction, pre-distortion or dispersion compensation were not employed in the experiments.

The encryption method is straightforward and does not need any additional bandwidth. Furthermore, it can be implemented in software, as an algorithm in digital signal processing or as a hardware solution. Therefore, it can very easily be implemented in the already existing and future network infrastructures for optical, THz or wireless systems with data rates in the Tbit/s range.

6. Line 190? the authors described that keyfuctions can be changed by phase modulation. Is it trivial to do in real time/ in-service?

Already the phase modulation of a single frequency line would further improve the security and the higher the number of lines that are independently phase modulated, the more difficult an attack would become. For a high-speed phase modulation (with up to the data rate of the signal transmission itself), a hardware phase modulator (for the single line, or several modulators for more lines) have to be used in the transmitter and receiver. The sequence and the start of the sequence for modulation have to be known by the transmitter and receiver. If the speed of the modulation is lower, it can be done by software or digital signal processing in the baseband.

For sharing the information of the phase modulation between transmitter and receiver we have several ideas, which would exceed the scope of the paper. The simplest is to exchange a predefined sequence between transmitter and receiver by a secure channel. Another possibility is to use a pre-defined open signal which is available and equal at the transmitter and receiver side. This can be a broadcast radio or TV signal, any signal from any available network, or noise from a source which can be detected and is equal at both sides.

7. Line 92: The agility in terms of keyfunctions, bandwidth and central frequency is constrained by the need of a secure channel. Can authors comment on it? How it compares with other techniques like steganography, spread spectrum and QKD etc?

Please see also answer 4, reviewer #2.

Yes, if the key, bandwidth, center frequency and key modulation are known by the attacker, the information in the channel is no longer secure. However, the parameters for the encryption have a low bandwidth and the secure channel has to exist for a short time only, whenever the parameters are changed. There are a lot of suggestions for secure information exchange, like QKD, post quantum cryptography or even hardware security primitives. The main difference of our method to other techniques is that very high bandwidth signals can be processed and that (besides the key parameter information) no additional redundancy has to be transmitted. QKD is secure in itself and does not need an additional secure channel. However, the bandwidth, transmission distance and data rate are quite low. The security of steganography or spread spectrum [10] may be based on the fact that the transmission is buried under noise and that a possible attacker does not see when and exactly in which spectral region a transmission happens. If so, the information about where and when has to be exchanged between the transmitter and receiver, which requires a secure channel too. If further measures, like a keyfunction for the spreading code are used, this information needs as well a secure channel. But the main disadvantage of

these techniques compared to our method is that much higher bandwidths are needed for the transmission. To highlight this better, we have included:

The keyfunction may be changed with a speed up to the symbol rate of the signal by a phase modulation, as well the frequency spacing of the flexible carrier source and symbol rate of the single signals might be changed during transmission. However, at every time the keyfunction including bandwidth, as well as center frequency of the signal must be known by the receiver. Therefore, a secure channel is required for the transmission of this information. Since the parameters for the encryption have a low bandwidth and the secure channel has to exist for a short time only, secure information exchange via QKD, post quantum cryptography or even hardware security primitives can be considered for this.

8. In case of low bandwidth signals, authors suggest filling the rectangular spectrum with noise or fake signals. Wouldn't it make transmission system spectrally inefficient by reducing the overall spectral efficiency? What about using flexible carrier source and baud rates to adjust the bandwidth according to signal bandwidth?

We thank the reviewer for this suggestion. Yes, additional noise and fake signals will make the transmission spectrally inefficient. The main advantage of our method is that we can transmit with the maximum possible symbol rate and a very high spectral efficiency. For the usage of flexible carrier sources and the adjustment to the signal bandwidths, please see answer #2. Regarding the noise and fake signals, we have changed the sentence to:

If only a single low-bandwidth signal has to be encrypted, the security of the transmission may be further enhanced by filling a broadband rectangular spectrum with additional noise or with encrypted fake-signals.

9. What about using a shared/ identical comb source (flexible carrier source) like the one in trasmitter as an LO in the receiver for CD?

We are not completely sure if we really have understood the suggestion. The transmitter and receiver are independent and at different locations. So, we cannot use the same comb source at both positions. However, we can use two independent comb sources with identical properties. We have changed Fig.3 (c) accordingly.

Reviewer #4 (Remarks to the Author):

In this manuscript, authors claim that they have developed a signal theory based encryption method in the physical layer to ensure security of fiber and wireless communication systems.

After carefully reading the manuscript, I think the manuscript is not suitable for publication.

First of all, the motivation of this work is doubtful. To ensure the security of data, current communication systems implement digital data encryption at higher layer of OSI stack -- for a good reason. Not all communication data needs to be encrypted, and not all data that needs encryption requires the same level of security and encryption. When high level of security is needed, more sophisticated encryption scheme can be used (and is available) against hostile attack. It would be a huge waste of resource to implement encryption or other security measures at physical layer for all data transmitted, which is what this manuscript suggests.

We completely agree with the reviewer that there are different possibilities, layers and needs for encryption. We are very sorry if we have given the impression that all data needs a physical layer encryption. Our technique can be used when needed, and only for the data that needs it. In a DWDM system a single channel can be en- and decrypted with the keyfunction, for instance, whereas all the other channels will stay completely unaffected. In an already existing transmission system this only requires a software update at the transmitter and receiver. Only if very broadband super-signals are to be transmitted and encrypted, additional hardware like a flexible carrier source is necessary. However, this holds for all kinds of super-signal transmission, if they are encrypted or not.

To make clear that the paper proposes just one technique for secure communications which might be used in next generation communication systems if needed, we have changed the title of the paper to: "Signal Theory Based Encryption of Faster-than-Nyquist Super-Signals for Next-Generation Fiber and Wireless Systems"

Additionally, we have changed the sentence:

Existing and future communication networks require an easy, on-the-fly, adaptable solution along with the compatibility to handle Tbit/s transmission rates that can be en- and decrypted with low-power, low-footprint integrated devices.

To:

Secure current and future communication requires an easy, on-the-fly, adaptable solution along with the compatibility to handle Tbit/s transmission rates that can be en- and decrypted with low-power, low-footprint integrated devices.

Secondly, the use of the term encryption in this manuscript is unorthodox. We normally refer to encryption as the digital data encryption. The idea suggested by the authors is what we normally call physical layer security, but we rarely refer to it as encryption. Physical layer security methods have been proposed for very specific applications such as in electronic warfare scenarios, they are not suitable for other or more general applications.

We have used the term encryption since we map the signal with the use of a keyfunction to a noise-like signal and use the same key for recovering the information from the noise. We think that with the presented experimental and simulation results we have shown that our method is suitable for many different applications.

Finally and most importantly, the authors fail to clearly describe their method and the uniqueness and advantages of their scheme over existing physical layer security schemes. The description given in the first paragraph of the section "Principle of en- and decryption" is very vague.

In the new version of the paper we have changed the description and hope that it can be understood better now.

The claim of a 252.4 GHz super-signal with a 7% higher symbol rate than given by the Nyquist limit is simply not possible -- did the authors find a way to break the Nyquist sampling theorem?

We do not break the sampling theorem. However, the Nyquist limit (which we break) defines the maximum possible symbol rate that can be transmitted error-free in a given channel bandwidth. So, breaking the Nyquist limit in first approximation simply means to have more errors. As long as the bit error rate is below the forward error correction limit (around 10^{-3}), the errors can be compensated. There are many papers about faster than Nyquist transmission in the literature. See for instance:

J. E. Mazo, "Faster-than-Nyquist signaling," in *The Bell System Technical Journal*, vol. 54, no. 8, pp. 1451-1462, Oct. 1975, doi: 10.1002/j.1538-7305.1975.tb02043.x.

T. Ishihara, S. Sugiura and L. Hanzo, "The Evolution of Faster-Than-Nyquist Signaling," in *IEEE Access*, vol. 9, pp. 86535-86564, 2021, doi: 10.1109/ACCESS.2021.3088997.

M. Ganji, X. Zou and H. Jafarkhani, "On the Capacity of Faster Than Nyquist Signaling," in *IEEE Communications Letters*, vol. 24, no. 6, pp. 1197-1201, June 2020, doi: 10.1109/LCOMM.2020.2980263.

From what I can gather in the description of the proposed method, it is no different than applying a frequency domain scrambler with a key function, which is nothing new and has been proposed by other works already.

We are very sorry if our description of the method has not been clear enough. In the new version of the paper we have tried to improve that. In a frequency domain scrambler, a fast Fourier transform (FFT) and an inverse FFT may be used to scramble single frequencies the signal consists of. This may be quite simple with speech and other low bandwidths signals, but for high-bandwidth signals the bandwidth of the required digital signal processing would become too low. In our method we are using sinc-pulse sequences (SPS), which are orthogonal to each other. As long as the sampling theorem is not violated, any number N of SPS, each of which a rectangular frequency comb with $N+1$ frequency lines in the spectrum, can be

used to define the signal error-free. The key function destroys the orthogonality between the SPS, which is then restored at the receiver side.

Sincerely,

A. Venugopalan on behalf of all authors

References:

- [1] Crockett, B., Romero Cortés, L., Konatham, S. R., & Azaña, J. (2021). Full recovery of ultrafast waveforms lost under noise. *Nature Communications*, 12(1), 2402.
- [2] Frascella, G., Agne, S., Khalili, F. Y., & Chekhova, M. V. (2021). Overcoming detection loss and noise in squeezing-based optical sensing. *npj Quantum Information*, 7(1), 72.
- [3] J. Zhou, Y. Qiao, Z. Yang, Q. Cheng, Q. Wang, M. Guo, and X. Tang, "Capacity limit for faster-than-Nyquist non-orthogonal frequency-division multiplexing signaling," *Scientific reports*, vol. 7, no. 1, p. 3380, 2017.
- [4] Mandalawi, Y., Meier, J., Singh, K., Hosni, M. I., De, S., & Schneider, T. (2023). Analysis of bandwidth reduction and resolution improvement for photonics-assisted ADC. *Journal of Lightwave Technology*, 41(19), 6225-6234.
- [5] He, Yang, et al. "High-speed tunable microwave-rate soliton microcomb." *Nature Communications* 14.1 (2023): 3467.
- [6] Sax, R. *et al.* High-speed integrated QKD system. *Photonics Res.* **11**, 1007–1014 (2023).
- [7] Alagic, Gorjan, et al. "Status report on the first round of the NIST post-quantum cryptography standardization process." (2019).
- [8] Mandalawi, Y., Singh, K., Hosni, M. I., Meier, J., & Schneider, T. (2023). High-Bandwidth Coherent OFDM-Nyquist-TDM Transceiver with Low-Bandwidth Electronics. *IEEE Access*, 11, 58244-58253.
- [9] Venugopalan, A., Mandal, P., Meier, J., Singh, K., & Schneider, T. (2024). Filterless reception of Terabit, Faster than Nyquist Superchannels with 4 GHz Electronics. *Journal of Lightwave Technology*.
- [10] Wohlgemuth, Eyal, et al. "Stealth and secured optical coherent transmission using a gain switched frequency comb and multi-homodyne coherent detection." *Optics Express* 29.24 (2021): 40462-40480.

Authors Response Letter (Manuscript ID: COMMS-24-0450-A)

Dear editor and reviewers,

Thank you for your decision letter dated 20.12.2024 to our research article entitled “Signal Theory Based Encryption of Faster-than-Nyquist Super-Signals for the Next-Generation Fiber and Wireless System”. We would like to thank the reviewers for their efforts and careful reviews of this paper. The suggestions offered by the reviewers have been immensely helpful.

-----Reviewer Comments-----

Reviewer #2 (Remarks to the Author):

The reviewer thanks the authors for their effort and for preparing the revision. My comments and concerns in the previous round have been adequately addressed and I have no further comments.

Reviewer #3 (Remarks to the Author):

The authors have made necessary revisions in the manuscript and provided satisfactory reply/explanation to suggestions and queries.

The manuscript can be accepted for publication.

Reviewer #4 (Remarks to the Author):

I am satisfied with the authors' modification of the paper according to reviewers' comments. Thanks to the authors' effort to clear some definitions and the overall organization of the paper. I think the manuscript is now ready for publication.

Sincerely,

A. Venugopalan on behalf of all authors